# Effects of aerosol in simulations of realistic shallow cumulus cloud fields in a large domain

George Spill[1], Philip Stier[1], Paul R. Field[2, 3], and Guy Dagan[1]

[1]Atmospheric, Oceanic and Planetary Physics, Department of Physics, University of Oxford, Oxford, UK
[2]Met Office, Exeter, UK
[3]Institute of Climate and Atmospheric Science, School of Earth and Environment, University of Leeds, Leeds, UK

**Correspondence:** George Spill (george.spill@physics.ox.ac.uk)

**Abstract.** Previous study of shallow convection has generally suffered from having to balance domain size with resolution, resulting in high resolution studies which do not capture large scale behaviour of the cloud fields. In this work we hope to go some way towards addressing this by carrying out cloud resolving simulations on large domains. Simulations of trade wind cumulus are carried out using the Met Office Unified Model (UM), based on a case study from the Rain In Cumulus over the Ocean (RICO) field campaign. The UM is run with a nested domain of 500km with 500m resolution, in order to capture the large scale behaviour of the cloud field, and with a double-moment interactive microphysics scheme. Simulations are run using baseline aerosol profiles based on observations from RICO, which are then perturbed. We find that the aerosol perturbations result in changes to the convective behaviour of the cloud field, with higher aerosol leading to an increase (decrease) in the number of deeper (shallower) clouds. However, despite this deepening, there is little increase in the frequency of higher rain rates. This is in contrast to the findings of previous work making use of idealised simulation setups. In further contrast, we find that increasing aerosol results in a persistent increase in domain mean liquid water path and decrease in precipitation, with little impact on cloud fraction.

## 1 Introduction

Shallow cumuli are the most common cloud type on Earth (Rossow and Schiffer, 1999; Sassen and Wang, 2008); they are ubiquitous throughout the trade winds, yet their behaviour is still poorly understood. These small, warm, shallow convective clouds have an important part in regulating the thermodynamics and dynamics of their environment; warming the cloud layer through condensation, transporting moisture to the inversion layer above, and cooling both the inversion and the sub-cloud layer through the evaporation of detraining cloud droplets and precipitation (Hartmann et al., 1992; Zhu and Bretherton, 2004; Neggers et al., 2007).

Trade wind shallow cumuli are of great interest in the context of a changing climate. In particular due to their coupling to circulation, as well as their radiative properties; reflecting shortwave radiation whilst emitting longwave radiation at a similar temperature to the surface due to their low, warm cloud tops. The myriad ways in which they interact with their environment means there is still much uncertainty in how they may respond to perturbations to the climate. Indeed, low cloud feedbacks are responsible for most of the uncertainty in climate sensitivity (Bony et al., 2004; Bony and Dufresne, 2005; Medeiros et al.,

2008; Vial et al., 2013; Boucher et al., 2013; Medeiros et al., 2015).

Aerosol particles in the atmosphere can act as cloud condensation nuclei (CCN) allowing the formation of cloud droplets (Köhler, 1936). Changes in aerosol concentration can therefore have significant impacts on the properties of clouds. For example, for a given liquid water content, an increase in CCN will lead to a greater number of smaller droplets. Smaller, more numerous droplets scatter more shortwave radiation back to space, and thus this results in an increase in the cloud albedo (Twomey, 1977). Additionally, the shift in the droplet size distribution may affect the formation of precipitation in shallow clouds by inhibiting the development of larger droplets (Albrecht, 1989).

Aerosol-induced changes in the precipitation efficiency of clouds can also lead to impacts on convection. Suppressed precipitation can result in increased condensation warming the lower part of the cloud layer, and increased evaporation of detraining droplets cooling the upper part. This destabilisation of the cloud layer can lead to an invigoration and deepening of the convection (Albrecht, 1993; Stevens and Feingold, 2009; Dagan et al., 2016; Sheffield et al., 2015).

Cloud fields may be affected in other ways; changing precipitation characteristics may affect the formation of cold pools, for example, which can have an impact on the development of new convection, and contribute to the mesoscale organisation of the field of shallow clouds (Seifert and Heus, 2013; Seigel, 2014; Seifert et al., 2015).

A number of studies (Xue et al., 2008; Jiang et al., 2010) have seen significant aerosol effects such as those described above. However, several others (van den Heever et al., 2011; Seifert et al., 2015) have also shown effects where parts of the system respond to perturbations in such a way as to offset the initial aerosol effect. Stevens and Feingold (2009) described these as buffering effects, and proposed possible buffers that may be relevant for cloud-aerosol interactions, including, for example, convective deepening and invigoration. They describe a mechanism for the deepening of shallow cumuli by increasing aerosol, whereby higher droplet numbers delay the onset of precipitation and increase evaporation at the cloud top. This destabilises the cloud layer, enabling greater vertical development of the cloud, which can then produce heavier rain, potentially compensating for the initial reduction in precipitation.

There have been a number of observational studies showing an invigoration effect on shallow clouds (Kaufman et al., 2005; Yuan et al., 2011; Koren et al., 2014), while modelling studies have shown seemingly conflicting results. Jiang and Feingold (2005) and Xue et al. (2008) both find that increasing aerosol actually suppresses convection in warm, shallow clouds, while Dagan et al. (2017) and Altaratz et al. (2014) argue for a 'turning point' between suppression and invigoration of convection, depending on local conditions and specific cloud properties. van den Heever et al. (2011) find that even within a cloud field the response varies; with shallower clouds being suppressed, and deeper clouds penetrating the trade inversion experiencing invigoration. A similar result is obtained by Seifert et al. (2015), who find a reduction in the number of small clouds due to an evaporative feedback from aerosol-suppression of precipitation. Both van den Heever et al. (2011) and Seifert et al. (2015) find that, though there are aerosol effects on cloud populations and properties such as rain rate, over a large area and after a long time these effects are minor. In contrast, Saleeby et al. (2015) find that a reduction in shallower cumuli and stratocumulus, along with an increase in deeper cumuli, leads to a reduction in domain accumulated precipitation with increased aerosol.

Much behaviour of convective clouds is constrained or driven by local conditions — heating, water budgets, or large scale subsidence for example — many of which may contribute to so-called buffering effects (Seifert et al., 2012), raising the possi-

bility that cloud responses to aerosol are regime- or regionally-dependent.

Despite much work on the subject, there is still a great deal of uncertainty and debate over the response of shallow convection to perturbations such as changes in aerosol (Tao et al., 2012). Typical modelling studies of shallow convection make use of high resolution large eddy simulations (LES), or cloud resolving models (CRM). These models explicitly resolve convection, but until recently have only been run on limited area domains, on the order of tens of km, due to computational limitations.

In this work we begin to extend the investigation of shallow convection by making use of the Met Office Unified Model's capabilities to run high resolution simulations on large domains, in order to study the effect of aerosol perturbations over entire cloud fields on spatial scales on the order of hundreds of km. Additionally, the use of a double moment cloud microphysics scheme, described below, allows aerosol concentration to be perturbed directly, rather than using cloud droplet number as a proxy. We aim to investigate the character of the response of shallow convection to aerosol perturbations in simulations of realistic weather systems, and whether and why this may differ from that seen in idealised simulations.

## 2   Model and case description

The Rain in Cumulus over the Ocean (Rauber et al., 2007) campaign was carried out over a period November 2004 - January 2005, in a region of the Trade winds in the western Atlantic off the Caribbean. This has been, and is, an ideal region for studies of shallow cumuli due to their prevalence, as well as the absence of upstream islands meaning that clouds observed here are likely to be highly representative examples of shallow cumuli.

Aircraft and ship-borne measurements across the campaign region were supported by ground-based systems as well as radiosondes. The aerosol profiles used in this work were based on measurements from one of the NSF/NCAR C-130Q campaign aircraft flights from 19/01/2005 (Stossmeister, 2008). Vertical profiles, shown in Fig. 1, of Aitken and accumulation mode aerosol number concentration were derived from a fit to this data, and allowed to decay exponentially with height (e-folding height = 1km) above 5km.

A global configuration of the UM vn10.8 (Walters et al., 2017), GA6.1, at resolution N768 (~25km x ~17km at midlatitudes) is run from operational analysis initial conditions, and used as a driving model to provide the lateral boundary conditions for a ~500km x ~500km nested region, centred on 17.5°N, 61.8°W. The nested region has a horizontal resolution of ~500m x ~500m, and a stretched vertical coordinate system with 70 levels below 40km. This nested configuration allows for the simulations to capture the transient features and forcing for the specific case, due to the open boundaries and driving global model. The resolution in the nested region is expected to resolve most of the relevant convection, and inspection of the simulations shows that this is indeed the case. A model time-step of 15s is used, with prognostic and diagnostic radiation time-steps of 900s and 300s. The simulations are initialised for 00:00 UTC 19 January 2005, and are run for 48 hours. The nested simulations are run without a parameterised convection scheme, and the operational microphysics scheme is replaced in favour of the double-moment Cloud AeroSol Interactive Microphysics (CASIM) scheme (Shipway and Hill; Grosvenor et al., 2017; Miltenberger et al., 2018). A number of size modes for insoluble and soluble aerosol are available, however we use only the soluble Aitken and accumulation modes, with the profiles shown in Fig. 1. These profiles are used to initialise the domain,

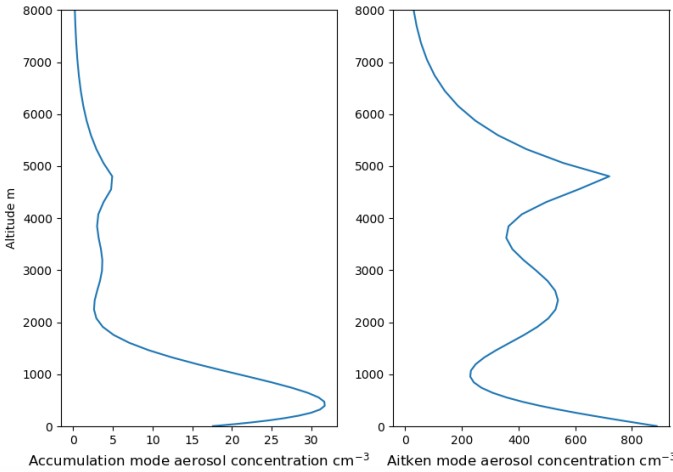

**Figure 1.** Vertical profiles of accumulation and Aitken mode aerosol concentration used in the baseline simulation.

and as lateral boundary conditions. Aerosol may be advected through the domain, but we do not include processing such as activation scavenging or precipitation washout. Here, CASIM is run with a sub-grid cloud fraction scheme based on that of Smith (1990), which parameterises the sub-grid variability in relative humidity. Its implementation in CASIM is described in Grosvenor et al. (2017). Additionally, we choose to apply the droplet activation scheme from Shipway (2015), rather than

CASIM's default scheme from Abdul-Razzak and Ghan (2000). This decision was based on the findings of a number of studies that the latter consistently underestimates the number of activated droplets for very high aerosol concentrations, and has too much competition for water vapour (Simpson et al., 2014; Connolly et al., 2014; Shipway, 2015). Shipway (2015) shows that this is particularly apparent for typical marine aerosol scenarios.

     Four simulations with different aerosol number concentrations were carried out: a baseline case UM_CASIM, and three with

the aerosol profiles perturbed by factors of 0.1, 10, and 100, labelled as UM_CASIM_0.1, UM_CASIM_10, and UM_CASIM_100.

## 3   Results and discussion

### 3.1   Structure and evolution of simulation

Figure 2 shows the evolution of a number of domain-average quantities over the simulation period for the baseline case, not including a 6 hour spin-up. Average profiles of liquid water potential temperature and specific humidity are shown in

Fig. 3, which compare well to those used as initial profiles used in the GEWEX Cloud System Study (GCSS) RICO model intercomparison study (vanZanten et al., 2011), as well as those shown in Nuijens et al. (2009), and those obtained from simulations such as in Seifert and Heus (2013). Qualitative visual inspection reveals the transient meteorological features and characteristics of the cloud field over the simulated period. Some example snapshots are shown in Fig. 4.

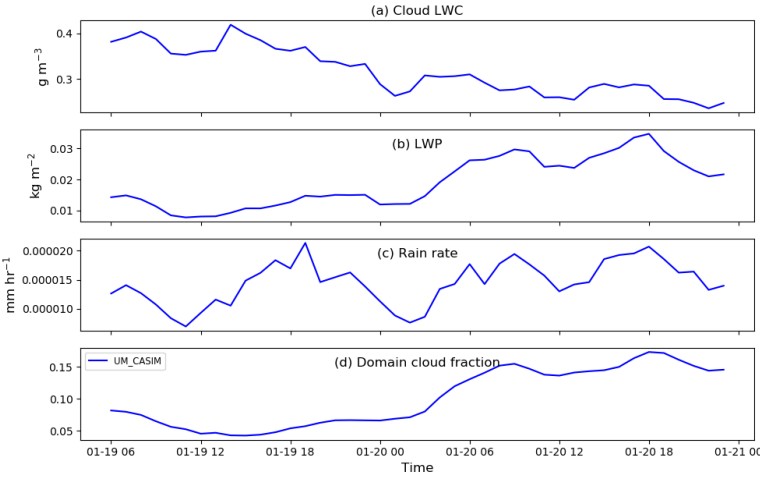

**Figure 2.** Time series of domain-average (a) cloud liquid water content (in-cloud only), (b) liquid water path, (c) rain rate, and (d) cloud fraction, all for the baseline case UM_CASIM. A liquid water content threshold of 0.01 g m$^{-3}$ is used to define a cloudy gridbox.

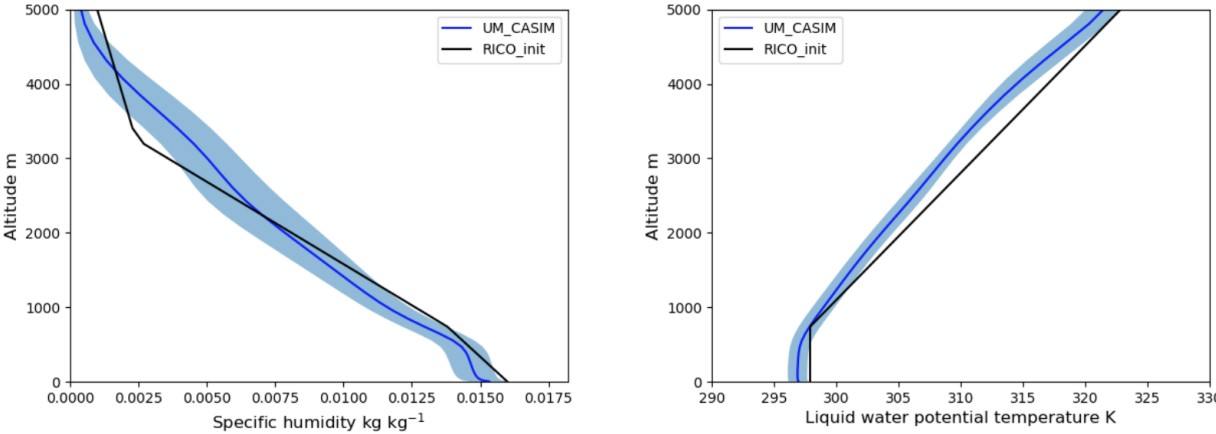

**Figure 3.** Domain-average vertical profiles of liquid water potential temperature and specific humidity for the baseline case, compared to the RICO initial setup, with the standard deviation of the baseline shaded.

## 3.2 Aerosol perturbations

In Fig. 5 time series of a number of domain average quantities, excluding an initial 6 hour spin-up period, show that even when considered across a large domain, there is a marked response to the aerosol perturbations.

Fig. 5(a) shows the domain average of in-cloud liquid water, with a liquid water threshold of 0.01 g m$^{-3}$ used to define a cloud. The cloud LWC and domain LWP both increase monotonically as the aerosol concentration is increased, while the rain

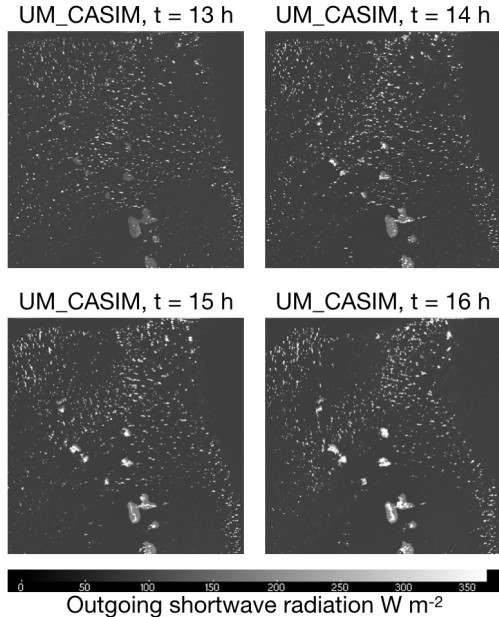

Outgoing shortwave radiation W m$^{-2}$

**Figure 4.** Snapshots of outgoing shortwave radiation showing the structure of the cloud field in the afternoon of the first day of the UM_CASIM simulation.

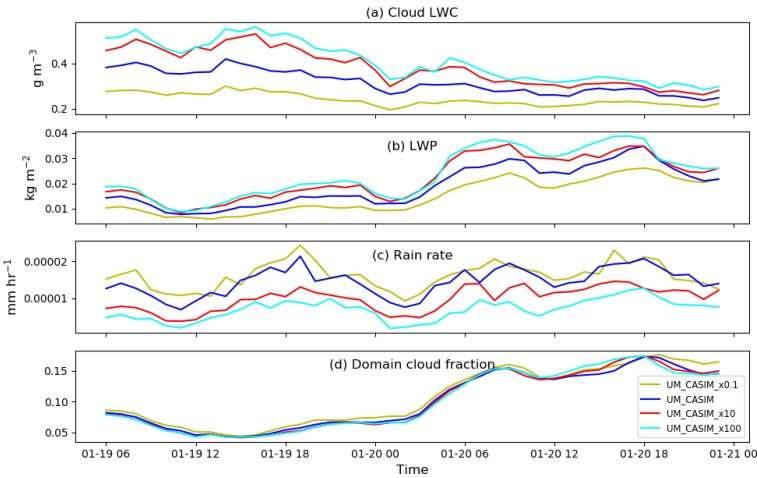

**Figure 5.** Time series of domain-average (a) cloud liquid water content (in-cloud only), (b) liquid water path (LWP), (c) rain rate, and (d) cloud fraction. A liquid water content threshold of 0.01 g m$^{-3}$ is used to define a cloudy gridbox.

rate decreases. Despite significant effects on other domain-wide parameters, there seems to be only a modest reduction of the domain-wide cloud fraction from aerosol perturbations.

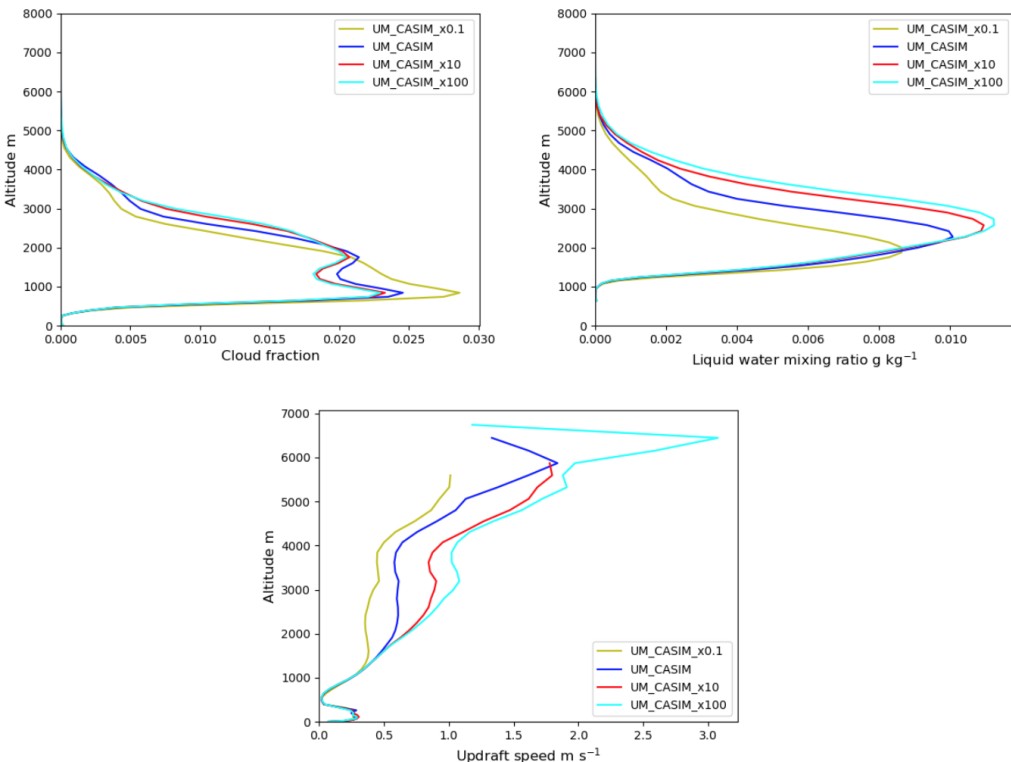

**Figure 6.** Vertical profiles of domain-average liquid water mixing ratio (top left) and cloud fraction (top right), calculated for all columns, and updraft speed (bottom), calculated for cloudy columns only.

Concurrently with the responses in average cloud LWC, in the domain average vertical profiles of liquid water mixing ratio in Fig. 6 we see similar trends. The peak in liquid water is increased, and is shifted to a higher altitude. Additionally, the liquid water mixing ratio becomes significantly greater at higher altitudes. The changes in these profiles indicate a deepening response to increasing aerosol. This may also be inferred from the profiles of cloud fraction (Fig. 6). As aerosol is increased, the cloud
5   fraction is reduced at lower altitudes, but increases at higher altitudes. Additionally, the lowest aerosol case, UM_CASIM_0.1, produces a cloud fraction profile which does not have the same pronounced double peaks seen in the other cases. An invigoration response is also evident in the profiles of updraft speed in Fig. 6. The updraft speeds show little change below 1.5km, however, there are marked responses above 1.5km, with updrafts increasing in strength with aerosol. This increase, along with smaller droplets under the higher aerosol conditions having smaller fall velocities, leads to more water being lifted higher in
10  the atmosphere (Koren et al., 2015), as can be seen in the vertical profiles of liquid water.

An increase in aerosol loading will generally lead to an increase in the cloud droplet number concentration. Under such an increase, combined with the increase in liquid water path, we would expect the cloud albedo to also increase. In Fig. 7 we show

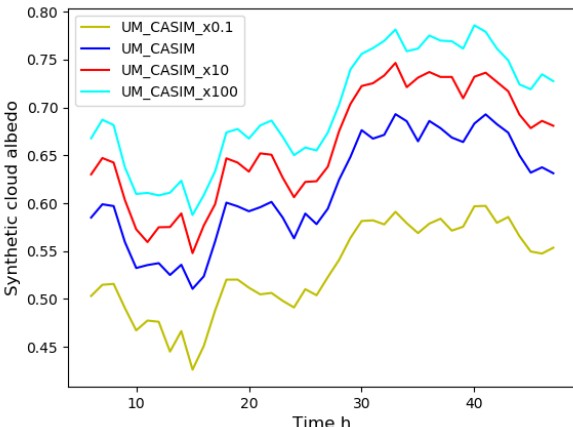

**Figure 7.** Time series of mean synthetic cloud albedo. This is calculated for cloudy columns only using an estimate of the cloud optical depth, as shown in equation 1.

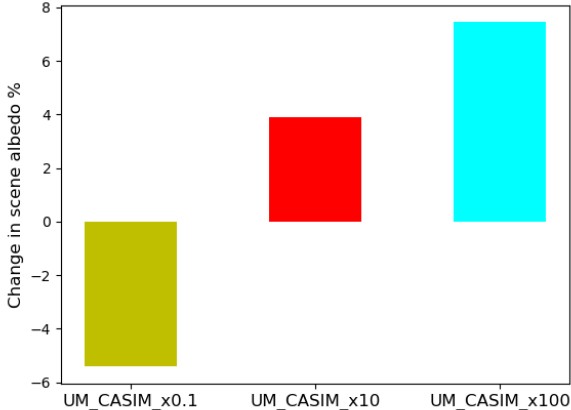

**Figure 8.** Percentage change in scene albedo for each of the perturbed aerosol simulations, relative to the baseline UM_CASIM case, calculated using time and domain mean synthetic cloud albedo and cloud fraction, as shown in equation 2.

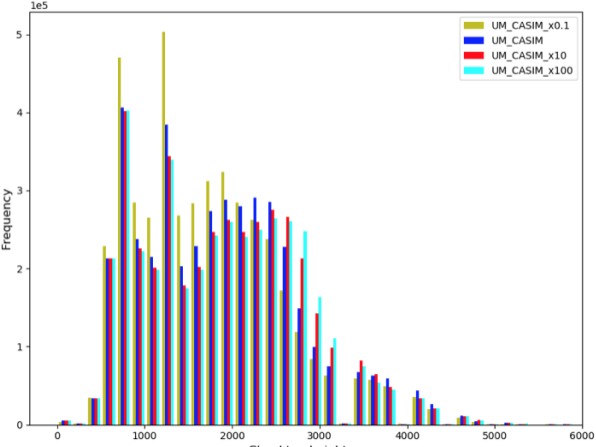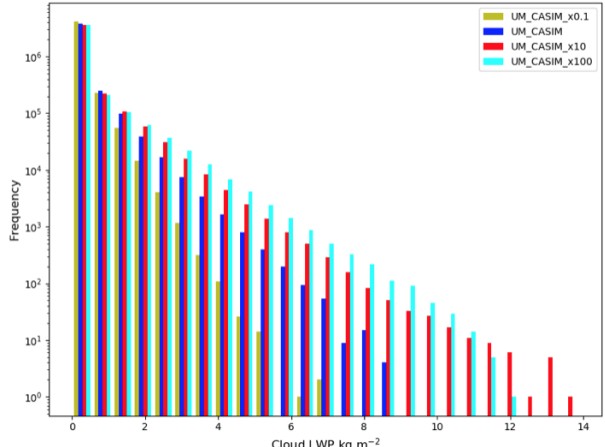

**Figure 9.** Histograms of cloud top height (left), calculated as the highest cloudy grid box in a cloudy column, and of total column liquid water path (right), calculated using only columns containing cloud.

the synthetic cloud albedo, calculated following Seifert and Heus (2013), as

$$A = \frac{\tau}{6.8 + \tau} \tag{1}$$

where $\tau$ is an estimate of optical depth given by $0.19 LWP^{5/6} N^{1/3}$, depending on the cloud liquid water path, LWP, and cloud droplet number concentration, N (Zhang et al., 2005). This shows that there is indeed a significant increase in the cloud albedo

with higher aerosol loads. This change is sufficient to lead to an increase in the domain-wide scene albedo, in spite of the slight reduction in cloud fraction with higher aerosol. This is shown in Fig. 8, where the scene albedo is calculated as

$$A_{\text{scene}} = CA + (1 - C)A_{\text{b}} \tag{2}$$

where $C$ is the cloud fraction, and $A_{\text{b}}$ is the background albedo. Following Seifert et al. (2015), we assume this to be the albedo of the sea surface at high zenith angles and set it to be 0.05.

The distributions of cloud top-height (CTH) shown in Fig. 9 also indicate a shift in the convective behaviour with aerosol perturbations; with increasing aerosol resulting in a suppression of the frequency of occurrence of clouds with lower CTHs, and an increase in the prevalence of higher CTHs.

The distributions of LWP in Fig. 9 also indicate a deepening response to aerosol; as the aerosol is increased, low LWPs become less frequent, while the tail of the distribution grows and extends to higher values.

The joint histograms of cloud top-height and liquid water path also shown in Fig. 10 give a clearer view of the effect; with higher aerosol concentrations come higher peak LWPs, indicating deeper clouds, as well as larger numbers of higher LWP clouds. However, it is also clear from these histograms that throughout all the simulations, the cloud fields are dominated in

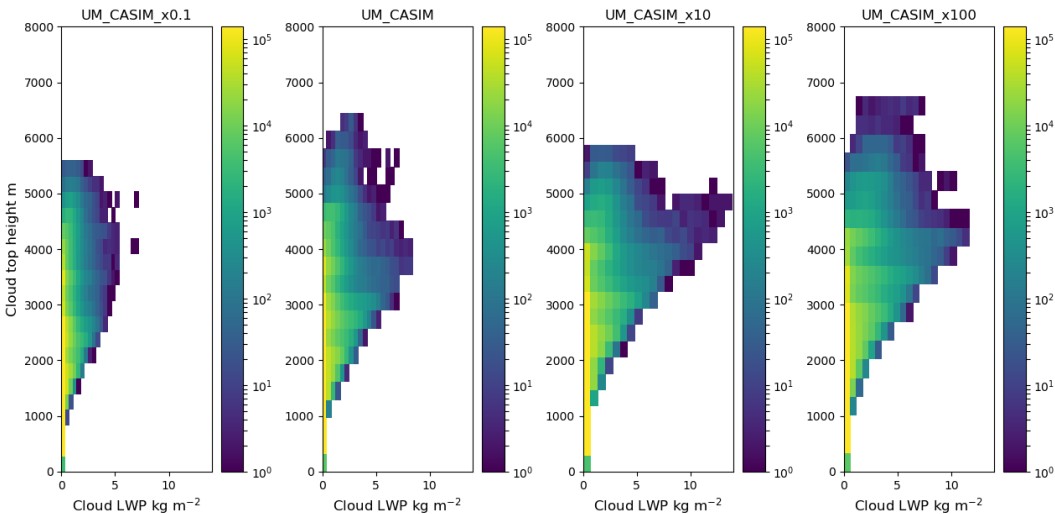

**Figure 10.** Joint histograms of liquid water path and cloud top-height for each simulation, with aerosol increasing from left to right along the figure.

terms of occurrence by the shallowest clouds with lower CTHs.

With the change in the convective behaviour of these clouds comes an effect on the precipitation. As shown in Fig. 11, lower aerosol concentrations result in higher frequencies of drizzle and lower rain rates, while these are suppressed for higher aerosol concentrations, as is the onset of precipitation. This effect is responsible for the reduction in the domain-average precipitation
5    in Fig. 2. There does not appear to be a consistent response in the frequency of the highest rain rates, however, due to the rarity of these events it is difficult to draw firm conclusions.

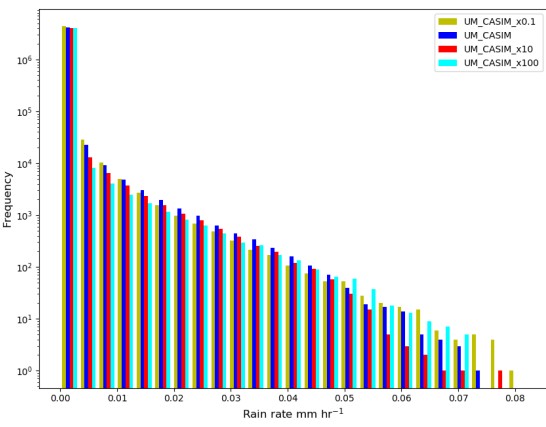

**Figure 11.** Histograms of rain rate, taken for all columns containing clouds.

We can attempt to gain a greater insight into the change in convection by inspecting the thermodynamic environment in which the clouds are developing. In Fig. 12 we see that the variation in the mean specific humidity and liquid water potential temperature is far less across the aerosol perturbations than the standard deviation in the baseline case. We can visualise the change in thermodynamic structure through the simulation in more detail using the plots shown in Fig. 13. Here we show the difference between the domain average temperature or specific humidity at a given time, and that at the beginning of the analysis period. We can see that though there are some differences between the simulations, they are not very large. The minor changes reflect the deepening of convection, but also demonstrate that the deepening and invigoration is not sufficient to significantly affect the thermodynamic structure in such a way as to promote further deepening.

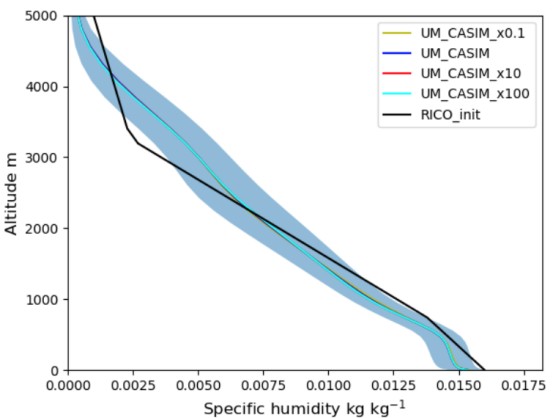 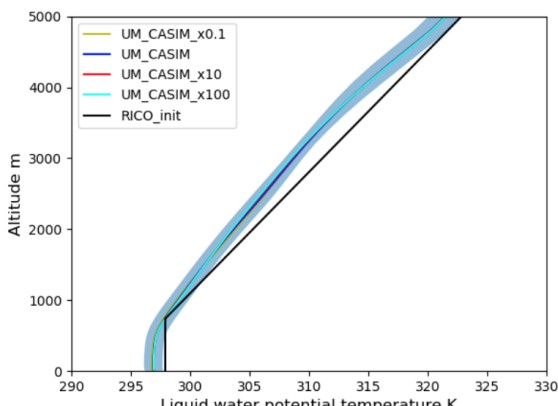

**Figure 12.** Average vertical profiles of liquid water potential temperature and specific humidity for each aerosol concentration, compared to the RICO initial setup as in Fig 2, with the standard deviation of the baseline case shaded.

## 4  Conclusions

We have presented results from a set of large domain simulations with perturbed aerosol loadings. Simulations based on particular, realistic days of the RICO field campaign were run using the Met Office Unified Model, in a 500kmx500km domain with 500m resolution, nested in a global driving model. Our findings show that for a large domain without periodic boundary conditions, with realistic synoptic weather subject to large scale forcing and energy and water budgets, changes in aerosol concentration can have significant effects. The impacts of aerosols on cloud microphysics are sufficient to result in persistent changes in the behaviour of the cloud field. We find that increasing aerosol suppresses the onset of precipitation, and leads to deepening and invigoration of convection. Increased aerosol loading results in a suppression of the shallow mode of convection, and invigoration of mid-level and deeper clouds. There is little change however in the updraft strength at low altitudes, in contrast to the substantially increased updraft speeds higher in the atmosphere. In spite of the convective deepening and invigoration, domain average precipitation is still reduced throughout the simulations, with little discernible change to the frequency

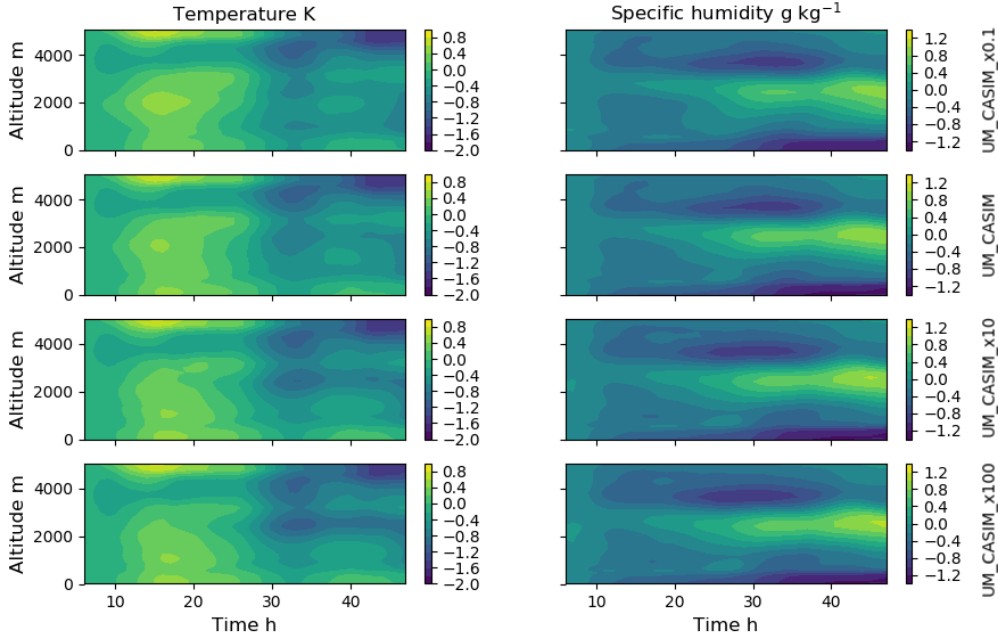

**Figure 13.** Hovmöller plots showing the temporal evolution of domain mean temperature (left) and specific humidity (right). These show the difference between the mean temperature or specific humidity at each time in the simulation and the first time point after the 6 hour spin-up. Each row is for a different simulation, with aerosol increasing from top to bottom down the figure.

of high rain rates. Examination of the thermodynamic structure of the simulations reveals that it is in fact highly resilient, and is not significantly affected by the changes to the cloud field.

Previous studies of the effects of aerosols on convection, which have made use of more idealised modelling setups, making use of prescribed forcings and periodic boundaries, have also found invigoration and deepening as a result of increasing aerosol, as well as suppression of the shallowest clouds. However, our results differ from these in several ways. van den Heever et al. (2011) and Seifert et al. (2015) both find similar suppression and invigoration effects on different parts of the cloud population, as well as on characteristics of the precipitation rate distribution. However they conclude that the domain-wide effect is minimal, with small impacts on domain average or equilibrium properties. This is in agreement with our own findings for cloud cover, but is in contrast to those for precipitation, where we find a persistent decrease in rain rates with higher aerosol. Further, while Seifert et al. (2015) find that reduced cloud cover with higher aerosol compensates for the Twomey effect to produce only a minor change in scene albedo in equilibrium conditions, we find that there is a clear increase in scene albedo with increasing aerosol. Lee et al. (2012) argue that differences in aerosol have significant effects on the thermodynamic environment and development of instability, which in turn affects development of the cloud field. Our findings, however, are of thermodynamic conditions which are not significantly affected by aerosol.

Additionally, Dagan et al. (2017) find that there exists an optimum aerosol loading for convective invigoration and deepening, above which the trend reverses and increasing aerosol leads to suppression. In contrast, we find monotonic deepening and invigoration. Our results here do not preclude the possibility of an optimum loading or turning point, although in this case it is likely that such a point would be far above realistic aerosol concentrations, given the perturbations we applied.

It is important to note that the standard picture of buffering of aerosol effects on shallow convection appears to require some equilibrium state of the cloud field. In idealised simulations this state is reached under different aerosol loadings by affecting the convective development and precipitation characteristics to varying degrees. However, in the real atmosphere with constantly varying cloud fields subject to large scale advection, no such equilibrium exists. Dagan et al. (2018) show that the characteristic timescale of shallow convective cloud fields is less than 12 hours, much less than the time required to reach an equilibrium state (Seifert et al., 2015). Here we have presented simulations of such a transient case, which suggest a quite different response; one in which cloud fields do not respond dramatically to restore an equilibrium, but instead are altered persistently, within the constraints of the transient thermodynamic conditions.

Given the apparent differences between idealised, limited area large eddy simulations, and those presented here, it seems clear that work is required to elucidate the sources of these differences. LES studies performed on large domains will be necessary, as well as direct comparison of idealised and realistic model setups. It will be important to discern which differences are due to the choice of model, and which are due to the idealised or realistic nature of the simulations. In future work we aim to make such a comparison through the use of idealised and realistic configurations of the same model. We hope that the simulations and results we have discussed here will provide a starting point for this direction of investigating anthropogenic perturbations to shallow cumuli, and ultimately the climate.

*Author contributions.* G. S. carried out the simulations and analyses presented. P. R. F. assisted with the simulations. P. S., G. D., and P. R. F. assisted with the design and interpretation of the analyses. G. S. prepared the manuscript with contributions from all co-authors.

*Competing interests.* The authors declare that they have no competing interests.

*Acknowledgements.* G. S. acknowledges funding from the Natural Environment Research Council with grant reference number 1796357, and from the UK Met Office, and the use of the Monsoon2 system, a collaborative facility supplied under the Joint Weather and Climate Research Programme, a strategic partnership between the Met Office and the Natural Environment Research Council. G. D. and P. S. acknowledge funding from the European Research Council project RECAP under the European Union's Horizon 2020 research and innovation program with grant agreement 724602. P.S. additionally acknowledges funding from the Natural Environment Research Council project NE/L01355X/1 (CLARIFY) and from and NE/P013406/1 (A-CURE).

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
