# Peer review of "Effects of aerosol in simulations of realistic shallow cumulus cloud fields in a large domain"

_Atmospheric Chemistry and Physics, 2019_

## Referee Comment (RC1) · Anonymous Referee #1 · 15 Jul 2019

Review of: "Effects of aerosol in simulations of realistic shallow cumulus cloud fields in a large domain"

Authors: George Spill, Philip Stier, Paul R. Field and Guy Dagan

Recommend minor revisions.

General comment: This manuscript is well-written and concise, and it discusses the impacts of aerosol loading on the albedo and cloud microphysics of trade wind cumuli over a large domain and at moderately high resolution. The results of aerosol effects appear consistent with many other studies, but also highlight differences that are still unclear to the broader community. Some of these differences may arise from variability in microphysics parameterizations rather than domain size or resolution as is empha-

sized here. With some minor changes, this paper should be ready for publication.

Specific comments:

1. Page 1, Lines 20-21: These sentences seem fragmented and/or run-on. Try clarifying your statements here starting with, "The coupling of trade wind. . .."

2. Page 2, Paragraph from Lines 23-32: You might also cite a paper by Saleeby et al. (2015, doi:10.1175/JAS-D-14-0153.1) that shows increased aerosol concentration leading to more rapid transition from stratocu to deeper cumuli via both enhanced droplet evaporation processes and invigoration. This study used an intermediate sized domain of 100km x 100km at 250m grid spacing.

3. Page 3, Line 17-20: Please include a figure that shows your baseline aerosol profile. Following initialization, can aerosols be advected, lofted, scavenged, etc?

4. Page 3: A few questions here on the model setup. Why does the model go up to 40km when the cloud analysis is focused below 7km? What's the vertical grid spacing in the cloud layer? Are you using open boundaries and nudging the boundaries with the operational analysis?

5. Page 3: Some would argue that 500m grid spacing is too coarse for simulating trade wind cumuli. There's a reason LES simulations are run for trade Cu at super high resolution. How big are your trade wind Cu and do you have enough grid points to adequate resolve these clouds? How many cloudy grid cells do you ultimately have in your analysis? Do you have enough cloudy area for a robust analysis? Just things to consider here with respect to resolution and cloud area. I feel the justification of this grid spacing for trade Cu needs to be a bit stronger.

6. Page 12, Lines 6-7: Perhaps it is not surprising that you do not see much change to the environment since the cloud fraction is so small. Perhaps the thermodynamic profiles surrounding the cloud field change, but this change gets averaged out when computing domain-wide properties. Please comment on this. Are there any changes

to sub-cloud evaporation and cold pools?

Figures: (Please make fonts clearer and larger in all figures)

Fig 7: Is the size of the colored box or histograms related to the range of values over the duration of the simulations?

Fig 8: Individual histograms are too small to see. Please make these larger and easy to discern.

Editorial comments: 1. Page 2, Lines 15-20: Too many uses of the phrase "a number of". 2. Page 3, Line 21: Error with the word "configuration". 3. Page 4, Line 7: Error with the word "Figure".

---

## Referee Comment (RC2) · Anonymous Referee #2 · 19 Jul 2019

The study investigates the impact of aerosol perturbations on shallow cumulus clouds by performing cloud resolving simulations on a larger domain than in previous work, while retaining relatively high spatial resolution. A general invigoration and deepening of convention with increasing aerosols is found, in line with previous work. Additionally, a number of key differences from more idealized simulations are identified that require further work by the community to be resolved. The paper is very well written and structured, and should be ready for publication with only minor revisions.

General comment: The "Results and discussion" section has lots of nice results and figures, but is quite descriptive and it would be good with some more discussion, e.g., comparison with previous work if possible, broader implications? Some of the former is included in the conclusions, but with limited discussion of why some of the differences

arise.

Specific comments: Pg. 2, line 8: perhaps something like "Aerosol-induced changes in precipitation efficacy . . ." would make the sentence easier to read.

Pg. 3, line 9: duplicate "the"

Pg. 3, line 18: It would be useful to know how these vertical profiles in the baseline look. Does the model simulate a mix of species? If so, how does this look?

Pg 4, line 7: Fig.ure -> Figure

Pg. 4, line 7-13: this paragraph introduces three figures over 7 lines with very little description or discussion of results. In particular, Figure 3 could need some elaboration to assist the reader in identifying these transient meteorological features. The information in Figure 1 is repeated in Figure 4 – is it really necessary to keep both?

Pg. 6, line 8: but the UM_CASIM_01 seems to make the low-altitude peak more pronounced? Suggest rephrasing to clarify.

Pg.6, lines 2-11: in this paragraph it would be useful to know more about the vertical aerosol profile.

Pg. 10, line 1-3: From "Shown in . . ."; something missing from this sentence. "As shown in. . ."?

Pg. 10, line 18: what "budgets"?

Pg. 12, line 8: please give some examples of what idealized means compared to this study, as the aerosol perturbations applied here could also be considered idealized.
* * *

---

## Author Comment (AC1) · 4 Sep 2019

Reply to the referees' comments on:

Effects of aerosol in simulations of realistic shallow cumulus cloud fields in a large domain

We would like to thank the referees for their constructive comments, which allowed us to make improvements to our paper. Please find below a response to each of the comments (shown in blue), with any alterations to the manuscript shown in italics.

Referee #1

General Comment: This manuscript is well-written and concise, and it discusses the

impacts of aerosol loading on the albedo and cloud microphysics of trade wind cumuli over a large domain and at moderately high resolution. The results of aerosol effects appear consistent with many other studies, but also highlight differences that are still unclear to the broader community. Some of these differences may arise from variability in microphysics parameterizations rather than domain size or resolution as is emphasized here. With some minor changes, this paper should be ready for publication.

Reply: We thank the referee again for this summary, and the further comments. We are pleased that the manuscript reads well.

Specific Comments:

1. Page 1, Lines 20-21: These sentences seem fragmented and/or run-on. Try clarifying your statements here starting with, "The coupling of trade wind. . .."

Reply: Following this comment, we have restructured the statements to make them clearer:

'Trade wind shallow cumuli are of great interest in the context of a changing climate. In particular due to their coupling to circulation, and their radiative effects; reflecting shortwave radiation whilst emitting longwave radiation at a similar temperature to the surface due to their low, warm cloud tops.'

2. Page 2, Paragraph from Lines 23-32: You might also cite a paper by Saleeby et al. (2015, doi:10.1175/JAS-D-14-0153.1) that shows increased aerosol concentration leading to more rapid transition from stratocu to deeper cumuli via both enhanced droplet evaporation processes and invigoration. This study used an intermediate sized domain of 100km x 100km at 250m grid spacing.

Reply: We would like to thank the referee for raising this to our attention. We have included a comment on the results of the mentioned paper, and how they compare to those already included:

'. . . Both van den Heever et al. (2011) and Seifert et al. (2015) find that, though

there are aerosol effects on cloud populations and properties such as rain rate, over a large area and after a long time these effects are minor. In contrast, Saleeby et al. (2015) find that a reduction in shallower cumuli and stratocumulus amount, along with an increase in deeper cumuli, leads to a reduction in domain accumulated precipitation with increased aerosol.'

3. Page 3, Line 17-20: Please include a figure that shows your baseline aerosol profile. Following initialization, can aerosols be advected, lofted, scavenged, etc?

Reply: Following this comment, we have included a figure of the profiles, which is provided with this response. We have further added to the manuscript: 'Vertical profiles, shown in Fig. 1, of Aitken and accumulation mode aerosol number concentration were derived from a fit to this data, and allowed to decay exponentially with height (e-folding height = 1km) above 5km.'

We have also added some more brief details of the microphysics scheme:

'A number of size modes for insoluble and soluble aerosol are available, however we use only the soluble Aitken and accumulation modes, with the profiles shown in Fig. 1. These profiles are used to initialise the domain, and as lateral boundary conditions. Aerosol may be advected through the domain, but we do not include cloud processing or aerosol removal by precipitation.'

4. Page 3: A few questions here on the model setup. Why does the model go up to 40km when the cloud analysis is focused below 7km? What's the vertical grid spacing in the cloud layer? Are you using open boundaries and nudging the boundaries with the operational analysis?

Reply: The model top of 40km was chosen in part because it is the standard for the Met Office Unified Model. Additionally, it means that the impact of ozone and water vapour in the upper atmosphere on the radiation is automatically considered. With a lower model top this would have to be computed anyway. The stretched vertical grid

has only 18 of the 70 levels are above 10km, while between 1-7km the spacing varies between ∼120-300m.

Regarding the boundaries, yes they are open and not periodic. The global configuration of the UM is run from operational analysis initial conditions, and provides the lateral boundary conditions for the nested domain. We have added to the model description in order to make this clearer:

'A global configuration of the UM vn10.8 (Walters et al., 2017), GA6.1, at resolution N768 (∼25km x ∼17km at midlatitudes) is run from operational analysis initial conditions, and used as a driving model to provide the lateral boundary conditions for a ∼500km x ∼500km nested region, centred on 17.5°N, 61.8°W. The nested region has a horizontal resolution of ∼500m x ∼500m, and a stretched vertical coordinate system with 70 levels below 40km. This nested configuration allows for the simulations to capture the transient features and forcing for the specific case, due to the open boundaries and driving global model.'

5. Page 3: Some would argue that 500m grid spacing is too coarse for simulating trade wind cumuli. There's a reason LES simulations are run for trade Cu at super high resolution. How big are your trade wind Cu and do you have enough grid points to adequate resolve these clouds? How many cloudy grid cells do you ultimately have in your analysis? Do you have enough cloudy area for a robust analysis? Just things to consider here with respect to resolution and cloud area. I feel the justification of this grid spacing for trade Cu needs to be a bit stronger.

Reply: We understand the referee's concerns over the choice of resolution. Part of the motivation for this work was to investigate the behaviour of cloud fields in a large domain, which added some computational limitations. We will investigate the sensitvity to model resolution in future work. However, most of the cumuli in our simulations cover more than a single grid cell, so while we may not capture as much detail as LES simulations, we are confident that we do capture the relevant the main behaviour.

There are approximately 4.1 million cloudy columns in total in the baseline simulation, which we feel provides an adequate number of clouds for the analysis.

6. Page 12, Lines 6-7: Perhaps it is not surprising that you do not see much change to the environment since the cloud fraction is so small. Perhaps the thermodynamic profiles surrounding the cloud field change, but this change gets averaged out when computing domain-wide properties. Please comment on this. Are there any changes to sub-cloud evaporation and cold pools?

Reply: Thank you for this comment, it is an interesting question. Local changes to the thermodynamics are possible, especially since there are changes in the convection and precipitation. However, we are aiming to show that the domain-wide conditions are resilient to these changes, and are determined more by the transient forcing of the particular case. The link between local changes and large scale tendencies will be a topic of further work, making use of a cloud tracking algorithm to analyse changes immediately surrounding the clouds. We are unable to comment directly on sub-cloud evaporation, due to limited model output. There do appear to be some differences between the simulations in the sub-cloud temperature, with the UM_CASIM_x10 simulation having a greater incidence of lower temperatures that may be associated with cold pools. However, there does not seem to be a clear or consistent response to the aerosol perturbations, or a clear link to the precipitation characteristics of the simulations. Additionally, there is little difference in the standard deviation of the near-surface buoyancy between the simulations, which has been shown to be associated with the prevalence of cold pools. There may be more local changes to cold pools, which is something we would investigate in the future along with the cloud tracking. It is also worth noting that, while the cloud fraction is indeed low, it is typical for this case, and is comparable with studies, referenced in the manuscript, such as vanZanten et al. 2011 and Seifert et al. 2015.

Figures:

We have adjusted the fonts in all figures.

Fig 7: Is the size of the colored box or histograms related to the range of values over the duration of the simulations?

Reply: The bars in figure 7 show the difference in scene albedo between each perturbed simulation and the baseline simulation, averaged across the duration of the simulations. They do not show any of the variation in the scene albedo over the course of the simulations.

Fig 8: Individual histograms are too small to see. Please make these larger and easy to discern.

Reply: We have made the histograms in figure 8 larger, as well as increasing the font size as in the other figures. We will work with the ACP production team to ensure good reproduction in the final paper.

Editorial comments:

1. Page 2, Lines 15-20: Too many uses of the phrase "a number of".

Reply: Thank you for this, in the revised manuscript the paragraph is now:

'A number of studies (Xue et al. 2008, Jiang et al. 2010) have seen significant aerosol effects such as those described above. However, several others (van den Heever et al. 2011, Seifert et al. 2015) have also shown effects where parts of the system respond to perturbations in such a way as to offset the initial aerosol effect. Stevens and Feingold (2009) described these as buffering effects, and proposed possible buffers that may be relevant for cloud-aerosol interactions, including, for example, convective deepening and invigoration. They describe a mechanism for the deepening of shallow cumuli by increasing aerosol, whereby higher droplet numbers delay the onset of precipitation and increase evaporation at the cloud top. This destabilises the cloud layer, enabling greater vertical development of the cloud, which can then produce heavier rain, potentially compensating for the initial reduction in precipitation.'

2. Page 3, Line 21: Error with the word "configuration".

Reply: This has been corrected.

3. Page 4, Line 7: Error with the word "Figure".

Reply: This has been corrected.

Referee #2

The study investigates the impact of aerosol perturbations on shallow cumulus clouds by performing cloud resolving simulations on a larger domain than in previous work, while retaining relatively high spatial resolution. A general invigoration and deepening of convention with increasing aerosols is found, in line with previous work. Additionally, a number of key differences from more idealized simulations are identified that require further work by the community to be resolved. The paper is very well written and structured, and should be ready for publication with only minor revisions.

Reply: We thank the referee for their comments and summary, identifying the key findings we wished to convey.

General comment: The "Results and discussion" section has lots of nice results and figures, but is quite descriptive and it would be good with some more discussion, e.g., comparison with previous work if possible, broader implications? Some of the former is included in the conclusions, but with limited discussion of why some of the differences arise.

Reply: We thank the referee for raising the point of developing the comparison with previous work, and investigating what causes the differences. At this stage we feel that commenting further than we have is a little too speculative, due to the differences in both the model and modelling setup employed by our work and that typically used to simulate shallow convection. However, we hope to be able to make a more direct comparison between the approaches in future work; through the use of 'idealised' and 'realistic' configurations of the same model. In order to make this clearer, we have

added to our conclusion:

'LES studies performed on large domains will be necessary, as well as direct comparison of idealised and realistic model setups. It will be important to discern which differences are due to model uncertainty, and which are due to the idealised or realistic nature of the simulations. In future work we aim to make such a comparison through the use of idealised and realistic configurations of the same model.'

Specific comments:

Pg. 2, line 8: perhaps something like "Aerosol-induced changes in precipitation efficacy . . ." would make the sentence easier to read.

Reply: Thank you, in the revised manuscript the sentence now reads:

'Aerosol-induced changes in the precipitation efficiency of clouds can also lead to impacts on convection.'

Pg. 3, line 9: duplicate "the"

Reply: This has been corrected.

Pg. 3, line 18: It would be useful to know how these vertical profiles in the baseline look. Does the model simulate a mix of species? If so, how does this look?

Reply: In the revised manuscript we have included a figure of the baseline aerosol profiles, which is provided with this response. The microphysics scheme is able to simulate a number of modes for soluble and insoluble aerosol. We have added some more detail to the description of the scheme:

'A number of size modes for insoluble and soluble aerosol are available, however we use only the soluble Aitken and accumulation modes, with the profiles shown in Fig. 1.

Pg 4, line 7: Fig.ure -> Figure

Reply: This has been corrected.

Pg. 4, line 7-13: this paragraph introduces three figures over 7 lines with very little description or discussion of results. In particular, Figure 3 could need some elaboration to assist the reader in identifying these transient meteorological features. The information in Figure 1 is repeated in Figure 4 – is it really necessary to keep both?

Reply: Thank you for raising this. In this instance and in reference to figure 3 (now figure 4), we are simply referring to the transient and varying nature of the cloud field. In the revised manuscript we have made this clearer:

'Qualitative visual inspection reveals the transient meteorological features and characteristics of the cloud field over the simulated period. Some example snapshots are shown in Fig. 4, where, for example, dense clusters and lines of clouds can be seen in different areas, as well as cloud free regions of varying sizes.'

We chose to separate the introduction of the structure of the baseline simulation from the comparison of the perturbed simulations. Although this does mean that the information in figure 1 (now figure 2 in the revised manuscript) is repeated, we feel it allows for easier inspection of the characteristics of the baseline.

Pg. 6, line 8: but the UM_CASIM_01 seems to make the low-altitude peak more pronounced? Suggest rephrasing to clarify.

Reply: Thank you. Indeed, as the reviewer pointed out, the low-altitude peak is more pronounced in UM_CASIM_0.1, however our intended meaning here was that the double-peak structure was less pronounced. This was not as clear as it could have been, and we have changed the sentence in the revised manuscript:

'Additionally, the lowest aerosol case, UM_CASIM_0.1, produces a cloud fraction profile which does not have the same pronounced double peaks seen in the other cases.'

Pg.6, lines 2-11: in this paragraph it would be useful to know more about the vertical aerosol profile.

Reply: As mentioned above, in the revised manuscript we have included a figure of the

vertical aerosol profiles, which we hope remedies this.

Pg. 10, line 1-3: From "Shown in . . ."; something missing from this sentence. "As shown in. . ."?

Reply: This has been changed to:

'As shown in Fig. 11, lower aerosol concentrations result in higher frequencies of drizzle and lower rain rates, while these are suppressed for higher aerosol concentrations, as is the onset of precipitation.'

Pg. 10, line 18: what "budgets"?

Reply: Thank you, this was not clear in the original manuscript. We are referring to the energy and water budgets due to our use of open boundaries and boundary conditions supplied by the global configuration of the Unified Model, in contrast to traditional LES. Following this comment, we have clarified our meaning in the revised manuscript:

'Our findings show that for a large domain without periodic boundary conditions, with realistic synoptic weather subject to large scale forcing and energy and water budgets, changes in aerosol concentration can have significant effects.'

We have also added a sentence to the model description in order to make the distinction between our simulations and typical periodic domains more clear:

'This nested configuration allows for the simulations to capture the transient features and forcing for the specific case, due to the open boundaries and driving global model.'

Pg. 12, line 8: please give some examples of what idealized means compared to this study, as the aerosol perturbations applied here could also be considered idealized.

Reply: Thank you for raising this, it is an important distinction to make. The aerosol perturbations we have applied are indeed idealised in that the profiles are simply multiplied by factors of ten, and are allowed to decay above 5km. However, in this instance we are referring to aspects of the modelling setup, rather than the experiments themselves. That is, the contrast between our use of open boundaries and a driving model (which we refer to as 'realistic' simulations), and the traditional LES approach using periodic boundaries and prescribed forcing (which we refer to as 'idealised' simulations). In addition to the changes mentioned above which describe in more detail some of the features of our model setup, we have amended this sentence to highlight the mentioned contrasts:

'Previous studies of the effects of aerosols on convection, which have made use of more idealised modelling setups, making use of prescribed forcings and periodic boundaries, have also found invigoration and deepening as a result of increasing aerosol, as well as suppression of the shallowest clouds.'
* * *
[Figure]

**Fig. 1.** Vertical profiles of accumulation and Aitken mode aerosol concentration used in the baseline simulation.